# Long-MS-Diff: Towards Generating Anatomically Plausible Lesion Progression in MS Imaging using Diffusion Models

**Prateek Mathur**[1,2] ID                              PRATEEK.MATHUR@UCD.IE
[1] *School of Computer Science, University College Dublin, Dublin, Ireland*
[2] *Insight Research Ireland Centre for Data Analytics, UCD, Dublin, Ireland*

**Brendan S. Kelly**[2,3] ID                        BREANDAN.KELLY@INSIGHT-CENTRE.ORG
[3] *School of Medicine, University College Dublin, Dublin, Ireland*

**Ronan P. Killeen**[3,4] ID                          RP.KILLEEN@ST-VINCENTS.IE
[4] *St. Vincent's University Hospital, Dublin, Ireland*

**Aonghus Lawlor**[1,2] ID                            AONGHUS.LAWLOR@UCD.IE

**Editors:** Accepted for publication at MIDL 2025

## Abstract

Monitoring lesion progression in Multiple Sclerosis (MS) is vital for assessing disease activity and guiding treatment decisions. However, the limited availability of annotated longitudinal MRI data presents a challenge for developing robust deep learning models. We introduce Long-MS-Diff, a conditional diffusion-based framework for generating anatomically plausible follow-up brain scans in MS. The model is conditioned on baseline images, lesion change masks, and scalar lesion-level features. To address the extreme sparsity of new lesions, we incorporate an auxiliary segmentation task and propose an adaptive weighted loss to balance anatomical reconstruction with lesion-specific fidelity. A radiologist assessment of synthetic scans confirms high image quality, anatomical plausibility, and lesion adherence. In downstream segmentation experiments, moderate augmentation with Long-MS-Diff improves performance, outperforming models trained on real data alone. These results highlight the value of controlled synthetic data in modelling disease progression and demonstrate the utility of diffusion-based generation in data-scarce clinical settings.

**Keywords:** Diffusion Models, Synthetic Medical Imaging, Multiple Sclerosis

## 1. Introduction

Multiple Sclerosis (MS) is a chronic neurodegenerative disease marked by the development of white matter lesions, visible on brain MRI (McNamara et al., 2017b,a). Accurately modelling lesion progression is essential for tracking disease activity and informing treatment decisions. However, the scarcity of annotated longitudinal data limits the development of models that capture the spatial and temporal dynamics of MS lesion evolution.

Generative models offer a promising direction for addressing data scarcity by synthesising realistic follow-up scans for data augmentation. In particular, diffusion models have shown strong performance in generating high-fidelity medical images (Kazerouni et al., 2023). Yet, they often prioritise image realism over explicit guidance (Preechakul et al., 2021), making it challenging to model subtle lesion progression. Structural causal models provide controllability but rely on balanced datasets of healthy and diseased cases, which are rarely available (Ribeiro et al., 2023).

In this paper, we propose Long-MS-Diff, a conditional diffusion framework for generating anatomically realistic follow-up scans in MS. The model incorporates scalar lesion-level features, an auxiliary segmentation task, and an adaptive weighted loss to achieve both anatomical consistency and lesion-specific fidelity. Unlike prior diffusion approaches, Long-MS-Diff is specifically designed to model sparse lesion progression through lesion-aware conditioning and dropout. We validate the framework through expert evaluation and demonstrate improved segmentation performance with synthetic follow-up data.

## 2. Methods

We propose Long-MS-Diff, a conditional diffusion framework built on improved denoising diffusion models (Nichol and Dhariwal, 2021), for generating anatomically plausible follow-up MRI scans in MS. The model is conditioned on a baseline image, a binary lesion change mask, and scalar lesion-level features (lesion load, lesion count, and a binary change flag), which are spatially broadcast and concatenated before being passed to a UNet-based diffusion backbone (Figure 1).

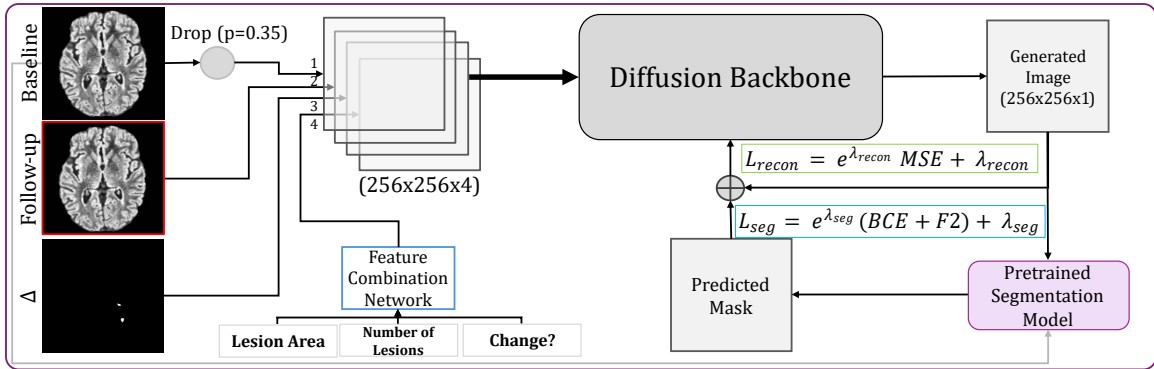

Figure 1: Long-MS-Diff model for generating anatomically plausible MS follow-up scans.

Recent advances in generative modelling have focused on conditional generation, where auxiliary guidance enforces semantic structure in the output (Konz et al., 2024; Saragih et al., 2024). In MS, a key challenge is the extremely low prevalence of new lesions, typically under 0.0013% of the image, which leads models to favour image realism over lesion fidelity. To address this, we introduce an auxiliary segmentation task, where a pretrained model segments new lesions in the generated scan, and the predicted mask is compared against ground truth using a segmentation loss ($L_{seg}$)

Diffusion models optimise for pixel-wise fidelity via a reconstruction loss $L_{\text{recon}}$, which can overpower lesion-specific supervision. We mitigate this with two strategies. First, we introduce an *adaptive weighted loss* that dynamically balances $L_{\text{recon}}$ and $L_{\text{seg}}$, ensuring both anatomical and lesion-level fidelity. Second, we implement a *conditioning dropout mechanism* by randomly replacing the baseline image with noise during training. This encourages the model to rely more on the lesion change mask and scalar features, promoting stronger lesion-aware generation.

## 3. Experiments and Results

We evaluate Long-MS-Diff on an internal longitudinal MS dataset comprising 170 patients, split into 110/30/30 for train/validation/test. Each patient has at least one baseline and one follow-up FLAIR MRI scan ($256 \times 256$), along with expert-annotated lesion change masks. The diffusion model is trained for approximately 18 hours[1]. A bi-temporal UNet serves as the pretrained segmentation model used for auxiliary supervision.

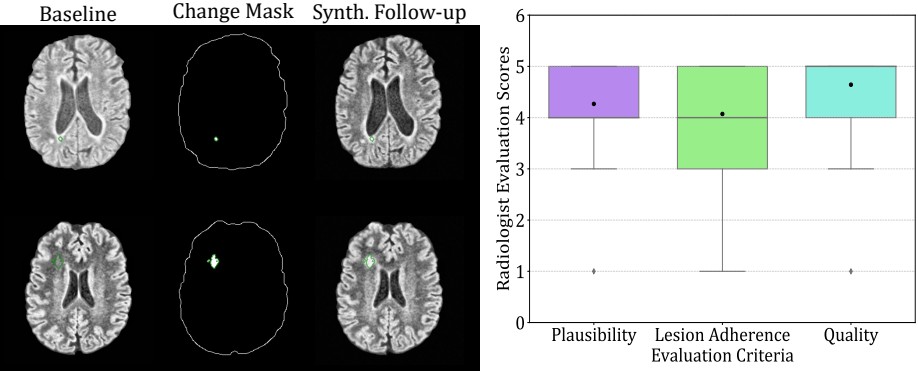

Figure 2: Examples of Generated Synthetic lesion progression (left). Radiologist evaluation of image quality, anatomical plausibility and lesion guidance adherence (right).

To qualitatively assess the realism of the generated scans, we synthesised 15,000 follow-up images using randomly sampled change masks. A random subset was reviewed by an expert radiologist, who rated each image on three criteria—overall image quality, anatomical plausibility, and lesion guidance adherence using a 5-point Likert scale. The images received high average scores—4.6 for quality, 4.2 for plausibility, and 4.0 for lesion adherence—indicating that the generated data is realistic and clinically meaningful.

Table 1: Performance of ViT (ResNet-50) trained with varying real-to-synthetic data.

| Real:Synthetic | Dice | Precision | Recall | Correct | False Pos. | False Neg. |
|---|---|---|---|---|---|---|
| 1:0 | 0.245 | 0.241 | 0.360 | 47 | 305 | 51 |
| 1:1 | 0.370 | **0.383** | 0.495 | 52 | **273** | 46 |
| 1:2 | **0.382** | 0.367 | **0.593** | **66** | 460 | **32** |
| 1:3 | 0.311 | 0.277 | 0.520 | 57 | 471 | 41 |

Additionally, we conduct a new lesion detection task using a Vision Transformer trained with different real-to-synthetic data ratios. Evaluation is carried out on real test sets following an established protocol (Kelly et al., 2024). Significant performance gains are observed (Table 1) with synthetic data augmentation peaking at a 1:2 ratio compared to no augmentation (1:0). A decline at 1:3 suggests overfitting to synthetic data and reduced generalisability. Despite modest performance, these results highlight the utility of Long-MS-Diff in generating clinically meaningful data for robust MS progression modelling.

## Acknowledgments

This research was supported by Science Foundation Ireland (SFI) under Grant Number SFI/12/RC/2289_P2 and the Irish Centre for High-End Computing (ICHEC).

---

1. Model was trained on 4×40GB NVIDIA A100 GPUs

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
