# OpenReview forum: "Long-MS-Diff: Towards Generating Anatomically Plausible Lesion Progression in MS Imaging using Diffusion Models"
_MIDL.io/2025/Short_Papers — MIDL 2025 - Short Papers_

### Official Review · Reviewer_m9yq · 2025-04-29

**Rating:** 4
**Confidence:** 5

**Summary:**

This paper proposes a conditional diffusion model to generate anatomically plausible follow-up MRI scans along with corresponding new multiple sclerosis (MS) lesion labels. Adaptive weighted loss and conditioning dropout techniques are applied to enhance the fidelity of new lesion synthesis. The synthetic MRIs are evaluated through both radiologists’ qualitative assessments and by incorporating them as additional training data to improve lesion segmentation performance.

**Strengths:**

•	The paper validates the fidelity of synthetic MRIs through both qualitative radiologist assessments and quantitative improvements in lesion segmentation accuracy.

**Weaknesses:**

•	It is unclear how the segmentation model would perform if trained solely on synthetic data.
•	It is not specified whether the downstream segmentation task focuses on new lesion detection or segmentation of all white matter lesions.

---

### Decision · Program_Chairs · 2025-05-01

Accept